# Comparison of the Fatty Acid and Triglyceride Profiles of Big Eye Tuna (*Thunnus obesus*), Atlantic salmon (*Salmo salar*) and Bighead Carp (*Aristichthysnobilis*) Heads

**DOI:** 10.3390/molecules24213983

**Published:** 2019-11-04

**Authors:** Jing Zhang, Ningping Tao, Yueliang Zhao, Xichang Wang, Mingfu Wang

**Affiliations:** 1College of Food Science and Technology, Shanghai Ocean University, Shanghai 201306, China; D170202038@st.shou.edu.cn (J.Z.); ylzhao@shou.edu.cn (Y.Z.); xcwang@shou.edu.cn (X.W.); 2Food and Nutritional Science Program, School of Biological Sciences, The University of Hong Kong, Hong Kong 999077, China

**Keywords:** marine fish, fresh water fish, DHA, triglyceride molecular species

## Abstract

Big eye tuna (*Thunnus obesus*), Atlantic salmon (*Salmo salar*) and bighead carp (*Aristichthys nobilis*) are three representative marine and fresh water fishes. In this study, the content of total lipids (TL), triglyceride (TG) fraction, and the fatty acid profiles in the corresponding fish heads were analyzed. Meanwhile, their complicated TG molecular species were further characterized. The results showed that TG was the major lipid in these three fish heads (60.58–86.69%). Compared with other two fish heads, big eye tuna head was the most abundant in polyunsaturated fatty acids, among which eicosapentaenoic acid (EPA) + docosahexaenoic acid (DHA) accounted for 64.29% and 32.77% in the TL and TG fraction, respectively. It is also worth noting that EPA+DHA/total fatty acid (TFA) value of TL and TG fraction from bighead carp head showed no significant difference with Atlantic salmon head, a typical marine fish. There were 146 TG molecules detected in big eye tuna head, 90 in Atlantic salmon and 87 in bighead carp heads. DHA or EPA accounted for 56.12%, 22.88%, and 5.46% of the total TG molecules in these three fish heads, respectively. According to principal component analysis, orthogonal projection to latent structures-discriminant analysis and the constructed heat map, the three samples could be completely differentiated based on their TG molecule fingerprints. This study is the first to compare marine and fresh water fish from the perspective of their heads’ fatty acid and TG molecule profiles.

## 1. Introduction

Omega-3 (ω-3) polyunsaturated fatty acids (PUFAs) are essential fatty acids which cannot be synthesized *de novo* by humans. Fish, flaxseed and some types of nuts such as walnuts are considered good sources of ω-3 PUFA [1]. Among PUFA, docosahexaenoic acid (DHA) and eicosapentaenoic acid (EPA) are known as effective antioxidative substances. It is also reported that sufficient intake of ω-3 PUFA shows neuroprotective activity and reduces the risk of cardiovascular diseases [2,3]. In addition, ω-3 fatty acids are reported to have the ability to change the structure and function of lipid microdomains (lipid rafts and caveolae) and play a distinct role in the promotion of health effects through modulation of membrane-signaling proteins (which are located in raft regions) [4].

As one type of representative marine fish, tuna contains a high amount of ω-3 PUFAs, typically EPA and DHA [5]. Salmon is also a good source of ω-3 PUFAs, and Larsen et al. found that PUFA of New Zealand King salmon fillets contained about 46.26% (EPA + DHA) [6]. Bighead carp (*Aristichthys nobilis*) is one of the four major fresh water fishes in China. Han and others [7] compared the fatty acid profiles of 15 major common fresh water fish muscles, the content of EPA + DHA in bighead carp was ranked as the fourth highest one. In addition, tuna, salmon and bighead carp are all widely consumed. The capture of big eye tuna (*Thunnus boesus*) was 472,934 tons in 2017, and the productions of Atlantic salmon (*Salmo salar*) and bighead carp in 2017 were 2,358,663 and 3,097,952 tons, respectively [8,9]. Their high content of EPA + DHA, and high production make these three fishes of great commercial potential.

High performance liquid chromatography (HPLC) with mass spectrum (MS) detection is an effective method to analysis triglyceride (TG) molecular species profiles. Non-aqueous reverse phase (NARP) mainly relies on carbon chain lengths and double bond numbers to separate TG molecules with different equivalent carbon number (ECN), which is defined as the total carbon number in the acyl chains minus two times the number of double bonds in fatty acyl chains. In general, a molecule with a short fatty acid chain will elute before a long-chain molecule, and a molecule with a high degree of unsaturation will elute before a highly saturated molecule [10,11]. As most natural TG molecules contain fatty acids in the range of C_16_–C_22_, the utilization of a C_18_ stationary phase will allow the maximum separation of TG molecules [12]. Electrospray ionization (ESI) is a soft ionization technique used for generating gas phase ions. In ESI analyses, a small amount of salt ions are often added to produce [M + NH_4_]^+^ or [Na]^+^ to perform the analysis [1,13,14,15]. Orbitrap and time of flight (TOF) mass spectrometers are two kinds of instruments of high resolution that are commonly used in metabolomics analysis. TOF instruments provide lipid profiling with high sensitivity, and Orbitrap instruments feature high resolution. Coupled with Orbitrap and TOF, Q Exactive provides the power of efficient data dependent acquisition, and acquires high resolution information of both precursors and fragments in the same analysis [16].

Fish heads are by-products generated during fish processing, which are normally used as fishmeal due to their low economic value Compared with other tissues fish heads might contain more abundant lipids aside from viscera. Meanwhile, the PUFA content is not low either [17,18]. Therefore, fish heads might a good source of lipids rich in PUFAs. In this study, the TG molecular species from big eye tuna, Atlantic salmon and bighead carp were determined and quantitatively analyzed using ultra high performance liquid chromatography (UHPLC)–Q E Orbitrap. Meanwhile, the corresponding lipid contents and fatty acid profile were also determined. This work will help to understand better of the lipid profiles from both marine and fresh water fishes, and provide a theoretical basis for the utilization of fish heads.

## 2. Results and Discussion

### 2.1. Lipids Contents and Fatty Acid Profile of Fish Heads

It was determined that the meat contents of big eye tuna, Atlantic salmon and bighead carp heads were 38.60 ± 4.01, 28.33 ± 3.22 and 43.05 ± 6.67%, respectively, while the total lipids contents in these three fish heads ranged from 5.61 to 15.67% of wet weight (Table 1), with the lipids contents in the two kinds of marine fish heads being markedly higher than that in big head carp heads.

The fatty acid profiles of the total lipids from the three kinds of fish heads were further analyzed (Appendix A). Their major fatty acids were consistent with data reported in some previous studies [18,19,20]. As shown in Figure 1A, bighead carp heads showed a significantly higher (*p* < 0.05) level of monounsaturated fatty acids (MUFAs) (61.81%) than Atlantic salmon (46.78%) or big eye tuna (14.33) heads, respectively, while the two kinds of marine fish heads contained significantly higher (*p* < 0.05) amounts of PUFAs than bighead carp heads (70.5%, 43.33% and 23.82%, respectively). Compared with MUFA and PUFA, saturated fatty acids (SFAs) represented the smallest fraction in all three types of fish head (15.16%, 9.37% and 14.37%).

Palmitic acid (16:0) was the major SFA in all three fish heads, accounting for 32.53%, 57.33% and 31.84% of the SFAs in the three species. The consumption of palmitic is related to the risk of cardiovascular disease. Palmitoleic acid (16:1), which found in human skin and decreases with age, was the major MUFA in all three fish heads, with 29.85%, 52.19% and 67.87%, respectively [21]. Large yellow croaker (*Larimichthys crocea*) is a typical marine fish species. Its muscle tissue contains 57.77% palmitic acid in the SFA fraction, and 22.88% palmitoleic acid in the MUFA fraction [22]. Meanwhile, tilapia is another major freshwater fish species. It was reported that 16:0 and 16:1 also are the predominant SFA and MUFA in tilapia heads, accounting for 68.8% and 13.78% of the SFA and MUFA, respectively [23].

EPA and DHA are two representative n-3 PUFAs that are widely found in marine triglycerides, phospholipids, ethyl esters and so on [24]. A large number of studies have demonstrated that dietary EPA and DHA could improve brain function, inhibit the activity of tumors and regulate lipid and glucose metabolism [25,26,27,28]. In this study, the amounts of EPA + DHA accounted for about 91.18, 34.42 and 53.18% in the PUFAs of the three kinds of fish heads. In contrast, the amount of EPA + DHA was 52.45% and 4.24% of PUFA in muscle of large yellow croaker and head of tilapia [22,23]. Meanwhile, the omega-3 indexes of big eye tuna, Atlantic salmon and bighead carp heads were 0.6, 0.14 and 0.12, respectively. It was reported that an omega-3 index cut-point value > 0.08 indicates a “low risk of coronary heart disease mortality” [29,30]. Therefore, it can be concluded that all three fish heads may serve as good potential sources of ω-3 PUFA. TGs represented 85.69, 60.58 and 73.39% of the total lipids in these three kinds of fish heads, and the assays of separated TGs reached 92–99%. As two representative marine fishes, the lipid contents in big eye tuna and Atlantic salmon heads were markedly higher than in bighead carp head (*p* < 0.05).

According to the distribution of SFA, MUFA and PUFA in the TG fraction (Figure 1B), the amount of UFA was higher than 60% in all three fish heads. Meanwhile, the TG fractions contained 32.77, 4.53 and 6.70% EPA + DHA, respectively. DHA was a major PUFA in the three fish heads, especially for big eye tuna, where it accounted for 81.78%.

It is worth noting that the (EPA + DHA)/TFA value in both total lipids and TG fraction of big tuna head was significantly higher than in the other two types of fish heads. Notably, bighead carp was showed no significant difference with Atlantic salmon, which is a typical marine fish. In terms of TG fraction, the bighead carp level was even higher than in Atlantic salmon (Figure 2). Therefore, bighead carp, a kind of fresh water fish species, can be a good source of EPA and DHA.

Besides that, the total lipids contained more EPA and DHA compared with the TG fraction in all three fish heads (*p* < 0.05). It could be speculated that the reason for this is that the total lipids is a mixture of different classes of lipids, including diglycerides, free fatty acids and phospholipids. Compared with the TG fraction, the other lipid components might contain more EPA and DHA.

### 2.2. Characterization of TG Molecular Species in Different Kinds of Fish Heads

Triglycerides (TG) are the major components in fish oil [31]. The WHO recommends that the consumption of 200–500 mg of fish oil is essential to satisfy the unsaturated fatty acid need in adults [24]. Besides that, EPA/DHA type of TG has higher bioavailability compared with the traditional ethyl ester (EE) EPA/DHA type [32]. The three fish heads were confirmed to be rich in PUFAs, especially EPA and DHA, in this study. If the fish heads could be used as a source of fish oil, it could represent a significant route for the high value utilization of byproducts. The physicochemical and nutritional properties of fish oil are determined mainly by the TG molecular species. Hence the TG molecule profile of the three fish heads was investigated. It is reported that EPA and DHA are not distributed randomly in a TG molecule, and the positional distribution of ω-3 PUFAs, especially EPA and DHA, will strongly influence TGs’ digestion and absorption [31]. 

Briefly, TGs are emulsified by intestinal peristalsis and bile salts, then hydrolyzed by enzymes in the intestinal lumen. The fatty acids at the sn-1/3 position are cleaved into free fatty acids. The hydrolyzed 2-monoglycerides and free fatty acids are absorbed by the intestinal epithelial cells and resynthesized into new triglyceride molecules, which are then transferred to the liver and adipose tissue via the lymphatic system and systemic circulation [33,34,35,36]. Hence, the identification of TG molecules may be helpful to reveal the beneficial effects of TG-fatty acids.

In this study, a UHPLC-Q E Orbitrap method was used to analyze the TG molecular species of big eye tuna, Atlantic salmon and bighead carp heads. High resolution mass spectrometry data-dependent analysis (HRMS-DDA) of positive ion modes were performed by Q E. The total ion chromatograms of the three kinds of fish heads in positive modes are shown in Figure 3. TG molecules could be selectively identified based on their loss of fatty acid chains.

The basic comparison of TG content was not enough for the identification of different sources of lipids. For more specific characterization, the comparison of distribution of TG molecule species in the three samples was necessary. Previously, there was no report about the comparison of marine and fresh water fishes from the perspective of TG composition. Based on our LC-MS/MS analysis, a total of 208 TG molecule species, ranging from *m*/*z* 766 to 1035, were identified in big eye tuna, Atlantic salmon and bighead carp heads, respectively. Among them, 146 TG molecule species were found in big eye tuna head, 90 TG molecule species were detected in Atlantic salmon and 87 in bighead carp heads, respectively. There were a totally of 28 TGs detected in all three kinds of fish heads (Table 2). A full list of the detected TG molecules is provided along with their absolute quantification data in the Appendix A. Furthermore, the contents of different TG molecular species were quantitatively analyzed with internal and external standards. The calibration curves are shown in Appendix A.

The accurate structure of an unknown TGs could be characterized according to its MS and MS/MS information. For example, based on the measured precursor ion *m*/*z* 946 ([M + NH_4_]^+^) of a unknown TG molecule, the product ions with *m*/*z* 601, 649, 656 were presented in its MS/MS spectrum. Then the fragments were matched with database using LipidSearch software v4.1.16 and identified as [DG (18:1/18:2) + NH_4_]^+^, [DG (18:1/22:6) + NH_4_]^+^ and [DG (18:2/22:6) + NH_4_]^+^, respectively (Figure 4). Hence, this TG could be speculated as TG (18:1/18:2/22:6). PUFAs are tend to located at sn-1/3 positions of TGs from marine mammals, and sn-2 position of TGs from fish oil [28,29]. Therefore, it could be speculated that 22:6 was located at sn-2 position, 18:1 and 18:2 were located at sn-1/3 position of this TG, respectively.

As listed in Appendix A, TG (16:0/18:1/18:1) (14067.04 ng/μL, 3.14%), TG (16:0/18:1/22:6) (24375.98 ng/μL, 5.44%), TG (16:0/22:6/22:6) (22494.45 ng/μL, 5.02%), and TG (18:1/22:6/22:6) (16613.31 ng/μL, 3.71%) were the dominant species in big eye tuna head. TG (18:1/18:1/18:1) (31636.9 ng/μL, 6.48%), TG (18:1/18:1/18:2) (28050.04 ng/μL, 5.74%), TG (18:1/18:2/18:2) (28646.7 ng/μL, 5.86%), TG (18:1/18:2/ 18:3) (26613 ng/μL, 5.45%), TG (18:3/18:2/18:2) (17218.12 ng/μL, 3.53%), and TG (20:1/18:1/18:1) (14686.86 ng/μL, 3.01%) were the dominant species in Atlantic salmon head. In bighead carp head, the dominant molecule species were TG (14:0/18:2/18:2) (10511.24 ng/μL, 4.2%), TG (16:0/16:0/18:2) (10752.92 ng/μL, 4.32%), TG (16:0/18:1/18:1) (9892.49 ng/μL, 3.96%) and TG (16:1/16:1/18:1) (9257.75 ng/μL, 3.70%).

Among the detected TG molecular species, there were 72 TG molecular species containing EPA or DHA in big eye tuna heads, accounting for 56.12% of the total TG molecules. Meanwhile, there were seven TG molecular species containing EPA or DHA in Atlantic salmon and bighead carp heads, accounting for 22.88 and 5.46% of total TG molecules (Figure 5). In terms of that, the three samples were significantly different with each other. The reason for this difference might be that big eye tuna and Atlantic salmon mainly feed on marine organisms rich in EPA and DHA, such as alewives, sea crab and shrimp, while bighead carp mainly feed on fresh water plankton, such as rotifers and cyanobacteria [37,38,39].

In the past, Zhang and others [20] examined the positional distribution of fatty acids of TG molecules using pancreatic lipase, and calculated the TG composition using a Visual Basic program. Their results showed that there were 30–40 TG molecular species in marine fish oil extracted from tuna and salmon. The dominant TGs in tuna oil were TG (20:5/16:0/18:0), TG (20:5/14:0/14:0), TG (16:0/22:6/22:6) and TG (16:0/18:1/22:6), which accounted for nearly 30% of the total TGs. Meanwhile, TG (16:0/16:0/18:0), TG (22:6/18:1/18:0) and TG (18:1/18:1/18:0) were the major molecular species in salmon oil, where they accounted for more than 42% in the total TG molecules. In our study, TG (16:0/22:6/22:6) and TG (16:0/18:1/22:6) were also found as the dominant species in big eye tuna head, while TG (18:1/18:1/18:0) was the main TG molecular species found in Atlantic salmon head.

As shown in Figure 6(a1), the TG molecules of the three kinds of fish heads were clearly differentiated from each other. The aggregate quality control (QC) value indicated that this UHPLC-Q E MS/MS method was able to distinguish the three samples based on their TG composition. The cumulative proportion for PC1 (which explained 74.8% of the variation) and PC2 (18%) was 92.8%. In order to improve the class separation and emphasize TG molecules responsible for sample separation, OPLS-DA was used by removing the structure noise affecting data matrix. As shown in Figure 6(a2–a4), with the removal of structure noise, there was a very effective separation among different kinds of fish heads with good predictive capability (R2X = 99.3, 90.4 and 97.3%, R2Y = 98.8, 97.3 and 98.7%, Q2 = 98.3, 88.3 and 98.1%).

Variable importance in the projection (VIP) score was calculated using a weighted sum of the squared correlations between the OPLS-DA components and the original variables following the literature [40]. In this study, the TG profiles of three kinds of fish heads were compared in pairs, and TG molecules with VIP>1 and *p* value < 0.05 were selected as discriminants able to classify the three kinds of fish heads. Top-50 differential TG molecules of the three samples were shown in Figure 6(b1–b3). There were two, five and 23 TG molecules containing EPA or DHA in the top-50 differential metabolites in big eye tuna heads, Atlantic salmon and bighead carp heads, respectively. As for TG molecule composition, there are obvious distinctions existing between marine and fresh water fish, and different marine fish species.

## 3. Materials and Methods

### 3.1. Materials

Half big eye tuna heads (number: 30; length: 28.0 ± 3.0 cm; weight: 1.55 ± 0.23 kg), Atlantic salmon heads (number: 30; length: 15.88 ± 1.20 cm; weight: 0.60 ± 0.1 kg) and bighead carp heads (number: 30; length: 18.50 ± 1.22 cm; weight: 0.95 ± 0.05 kg) were purchased from Xiang Xiang Food Co., Ltd. (Dalian, Liaoning, P.R. China). Originally, the big eye tuna were captured in the Pacific-Indian Ocean region in September, the farmed Atlantic salmon was taken the Danish Faroe Islands in October, and the farmed bighead carp was obtained from Qiandao Lake in the Chinese Zhejiang region in October.

The meat and bone of fish heads were divided using a scalpel, then the head meat was packaged at 30 g/bag after homogenization at a rate of 5000 r/min for 10 min using a homogenizer (JHBE-30A, Fanzhi, Shanghai, China). Then the divided samples were stored at −30 °C for a maximum of 4 weeks until use.

Organic solvents (including analytical and HPLC/gas chromatography (GC) grade), C19:0 and C19:0 FAME, and the mixture of 37 fatty acid methyl esters (FAME) were purchased from Shanghai ANPEL Scientific Instrument CO., Ltd. (Shanghai, P. R. China). According to the manufacturer, all the FAMEs in the mixture were of equal weight (i.e., 2.63% of the mixture). The TG standards used in this study were TG (18:1/18:1/18:0) and TG (16:0/18:0/16:0)-d5, all at ≥99% were purchased from Lordan (Solna, Sweden).

### 3.2. Sample Preparation

Total lipids were extracted according to the method of Folch and others [41]. Ten grams of head meat sample were mixed with 200 mL chloroform-methanol (2:1, *v*/*v*), and extracted for 24 h at 4 °C. The extracted solution and residue were then divided by filtration. Then, 30 mL of 0.9% NaCl solution was added into the extracted solution and kept at 4 °C for 8 h, the lower chloroform phase was collected, the chloroform was removed by rotary vacuum evaporator (RV10, IKA, Staufen, Germany) at 40 °C. The extracted total lipids were stored at −30 °C for a maximum of 1 week until use.

The TG fraction of fish heads was purified using silica gel (100–200 mesh, Merck, Darmstadt, Germany) column chromatography [42]. Briefly, the silica gel column was equilibrated with dichloromethane/*n*-hexane, 2:3 (*v*/*v*). Then the TG fraction was eluted using dichloromethane. The eluent was collected and the mobile phase was removed by rotary vacuum evaporator at 40 °C. The remaining eluted lipids were weighed on an analytical balance (Mettler-Toledo, Zurich, Switzerland), respectively. Then the remaining eluted lipids were stored at −30 °C for a maximum of 1 week until use.

### 3.3. TG Fraction Analysis

The separated TG fraction was determined using an Iatroscan MK-6S thin layer chromatography-flame ionization detection (TLC-FID) Analyzer (Iatron Inc., Tokyo, Japan), according to the method of Yin [43].

### 3.4. Determination of Fatty Acids Profiles

The total lipids and TG fraction of big eye tuna, Atlantic salmon and bighead carp heads meats were converted to methyl ester derivatives following the method of Zhang [44]. Briefly, 0.1 g of lipids and 100 μL of C19:0 internal standard (10 mg/mL) were added into 5 mL of methanolic-NaOH (0.5 mol/L). Then the mixture was in a condensing and concentrating equipment (HWS24, HongLang, Zhengzhou, Henan, P.R. China), heated at 100 °C for 10 min. After that, 3 mL boron trifluoride-methanol (14% in methanol) was added to the mixture at 100 °C and stirred for 3 min, followed by the addition of 2 mL n-hexane and holding at 100 °C for 2 min. Finally, 10 mL saturated NaCl solution was added to the mixture. The sample was cooled down to room temperature (24–27 °C), the upper n-hexane layer was collected using a 2 mL disposable syringe and purified with a nylon syringe filter (13 mm × 0.22 μm, ANPEL Inc., Shanghai, China) and stored in 2 mL thread screw neck vial with a septum (32 × 11.6 mm, ANPEL Inc.) for further analysis.

GC analysis was carried out to determine the fatty acid profiles. A gas chromatograph TRACE GC ULTRA (Thermo Fisher Inc., Waltham, MA. USA) equipped with an Agilent (Santa Clara, CA. USA) SP-2560 capillary column (100 m length × 250 μm internal diameter, 0.2 μm of film) and a flame ionization detector (Thermo Fisher Inc.) was used. The temperature of the column ramp was: the initial temperature was 70 °C, heated to 140 °C (20 °C/min), held for 1 min; then to 180 °C (4 °C/min), held for 1 min; then to 225 °C (3 °C/min), held for 30 min. The gasifying temperature was 250 °C. The flow rate of N2 was 1 mL/min. The injection volume was 1 μL, with a split ratio of 45:1. FAME were identified by comparison of their retention time with standard mixture. The contents of different fatty acids were determined using the area ratio of GC peak of internal standard C19:0 and different fatty acids tested. The specific calculation formula as follows:$$X_{i}=F_{i}\times \frac{A_{i}}{A_{C19:0}}\times \frac{C_{C19:0}\times V_{C19:0}\times 1.047}{m}\times 100\times F_{\mathrm{FAMEi}-\mathrm{FAi}}$$
where Xi is the contents of different fatty acids, mg/g; Fi is the response factor of each FAME; Ai is the peak area of each FAME in the sample; A19:0 is the peak area of the internal standard C19:0; CC19:0 is the concentration of C19:0, mg/mL; VC19:0 is the volume of the internal standard C19:0, mL; 1.047 is the transfer coefficient of C19:0 to C19:0 FAME; FFAMEi-FAi is the transfer coefficient of FAME for each fatty acids; m is the mass of total lipids, in g.

Fi was determined using the following expression:$$F_{i}=\frac{C_{\mathrm{Si}}\times A_{19:0}}{A_{\mathrm{Si}}\times C_{19:0}}$$
where C_Si_ is the concentration of each FAME in the mixed standard, mg/g; A_19:0_ is the peak area of C_19:0_ FAME standard; A_Si_ is the peak area of each FAME in the mixed standard; C_19:0_ is the concentration of C19:0 FAME standard, mg/g. Fatty acid composition was expressed as mg/g of total lipids.

### 3.5. Determination of TG Molecular Species

An UHPLC method was used following the method published by Xu [16]. Briefly, an ultimate 3000 UHPLC (Dionex) coupled with a Q Exactive MS system (Thermo, Waltham, MA. USA) was used to perform LC separation. A Cortecs C18 column (2.1 × 100 mm, Waters, Milford, MA, USA) was applied for analysis, the column chamber temperature was 40 °C. Mobile phase A was prepared by dissolving 0.77 g of ammonium acetate into 400 mL of HPLC-grade water, followed by adding 600 mL of HPLC-grade acetonitrile (pH~7). Mobile phase B was prepared by mixing 100 mL of acetonitrile with 900 mL isopropanol. The gradient was as followed: 0 min, 33% B; 2.5 min, 33% B; 5 min, 45% B; 6 min, 52% B; 9 min, 58% B; 12 min, 66% B; 15 min, 70% B; 19.5 min, 98% B; 29.5 min, 98% B; 30 min, 33% B; 35 min, 33% B. The flow rate was 0.25 mL/min.

The MS method followed the method published by Xu [16]. The detailed mass spectrometer parameters were as follows: spray voltage, 3.2 kV for positive and 2.8 kV for negative; capillary temperature, 320 °C; sheath gas flow rate (Arb), 35; aux gas flow rate (arb), 10; mass range (*m*/*z*), 240–2000 for positive and 200–2000 for negative; full MS resolution, 70,000; MS/MS resolution, 175,000; topN, 10; NCE, 15/30/45; duty cycle, 1.2 s.

The obtained TG fraction of 2.3 was diluted 80 times (*v*:*v*) using dichloromethane/methanol (2:1, *v*/*v*), the diluted TG was used to analyze. An absolute quantitative method using internal and external standards was carried out in this study. TG (16:0/18:0/16:0)-d5 was added to the samples as the internal standard at a concentration of 0.2 μg/mL. TG (18:/18:1/18:0) was used as external standard. Like the samples to be tested, 0.2 μg/mL of TG (16:0/18:0/16:0)-d5 was added as an external standard. In the calibration curve, Y was peak area ratio of internal standard and external standard, and X was the content of TG (18:/18:1/18:0) precursor ion. The absolute content of tested TG molecules could be realized using put the ratio of internal standard and tested TG molecular precursor ion into the calibration equation. Injection volume of both samples and external standard was 1 μL.

### 3.6. Data Analysis

Lipids were identified using LipidSearch software v4.1.16 (Thermo). Adducts of +NH4 were applied for positive mode search as ammonium acetate was used in mobile phases. Results were expressed as mean ± standard error of the mean. All statistical analyses were done using SPSS version 10.0 software (SPSS Institute, Inc., Chicago, IL, USA). *p* values < 0.05 were considered statistically significant. The raw MS data was exported into SIMCA-P (Umetrics, Malon, Sweden) to perform principal component analysis (PCA) and orthogonal projection to latent structures-discriminant analysis (OPLS-DA). Subsequently, heat maps were generated using R (http://www.r-project.org/).

## 4. Conclusions

Big eye tuna, Atlantic salmon and bighead carp are representative marine and fresh water fishes. TGs are major lipid class in all three kinds of fish heads. Compared with fresh water fish, marine fish heads, especially big eye tuna, contain higher contents of EPA and DHA. A total of 208 TG molecules were detected in these three kinds of fish heads, and the dominant TG molecules of each fish species head were identified. Based on unsupervised PCA and supervised OPLS-DA methods, the TG molecule compositions of the different fish species could be completely distinguished, and heat maps showed an obvious distinction between the marine and fresh water fish species.

## Figures and Tables

**Figure 1 molecules-24-03983-f001:**
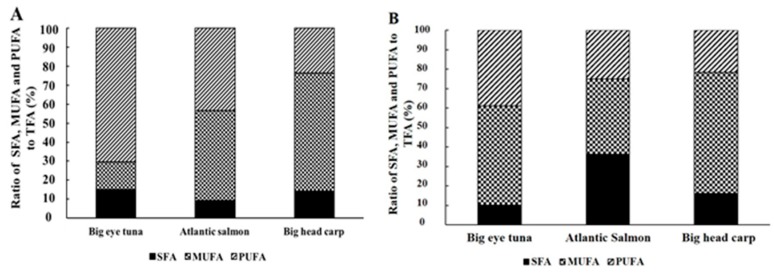
Ratio of SFA, MUFA, PUFA to TFA. (**A**) Total lipids, (**B**) TG fraction of the three kinds of fish heads.

**Figure 2 molecules-24-03983-f002:**
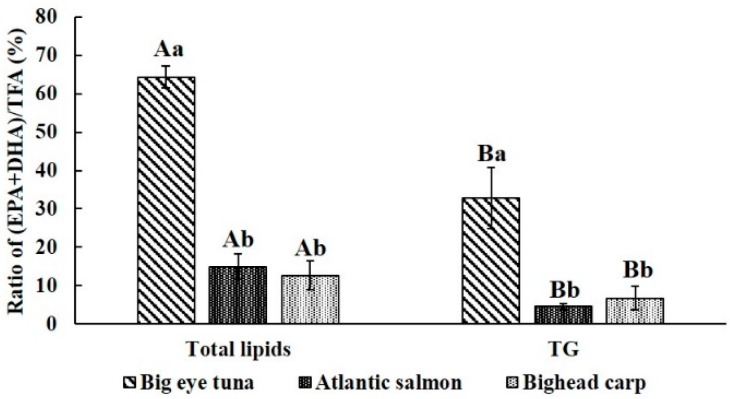
Ratio of (EPA + DHA) to TFA of total lipids and different classes of lipids. ^A,B^ means a significant difference of total lipids and TG fraction, ^a–c^ means a significant difference between the three different types of fish heads, *p* < 0.05.

**Figure 3 molecules-24-03983-f003:**
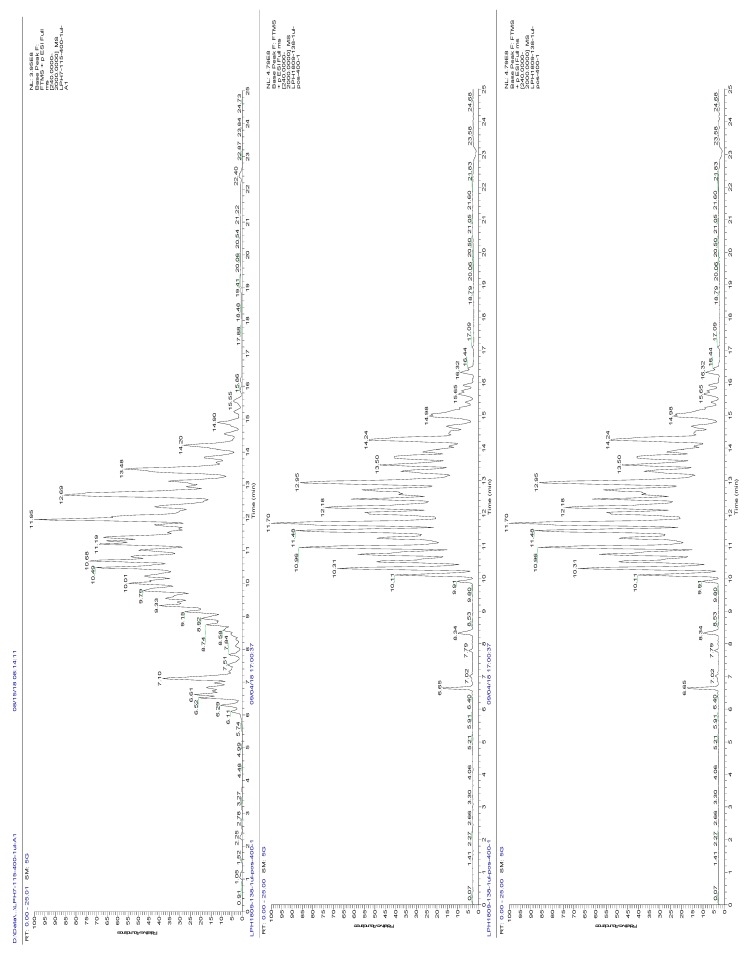
Total ion chromatogram of the three kinds of fish heads. Top, big eye tuna; middle, Atlantic salmon; bottom, bighead carp.

**Figure 4 molecules-24-03983-f004:**
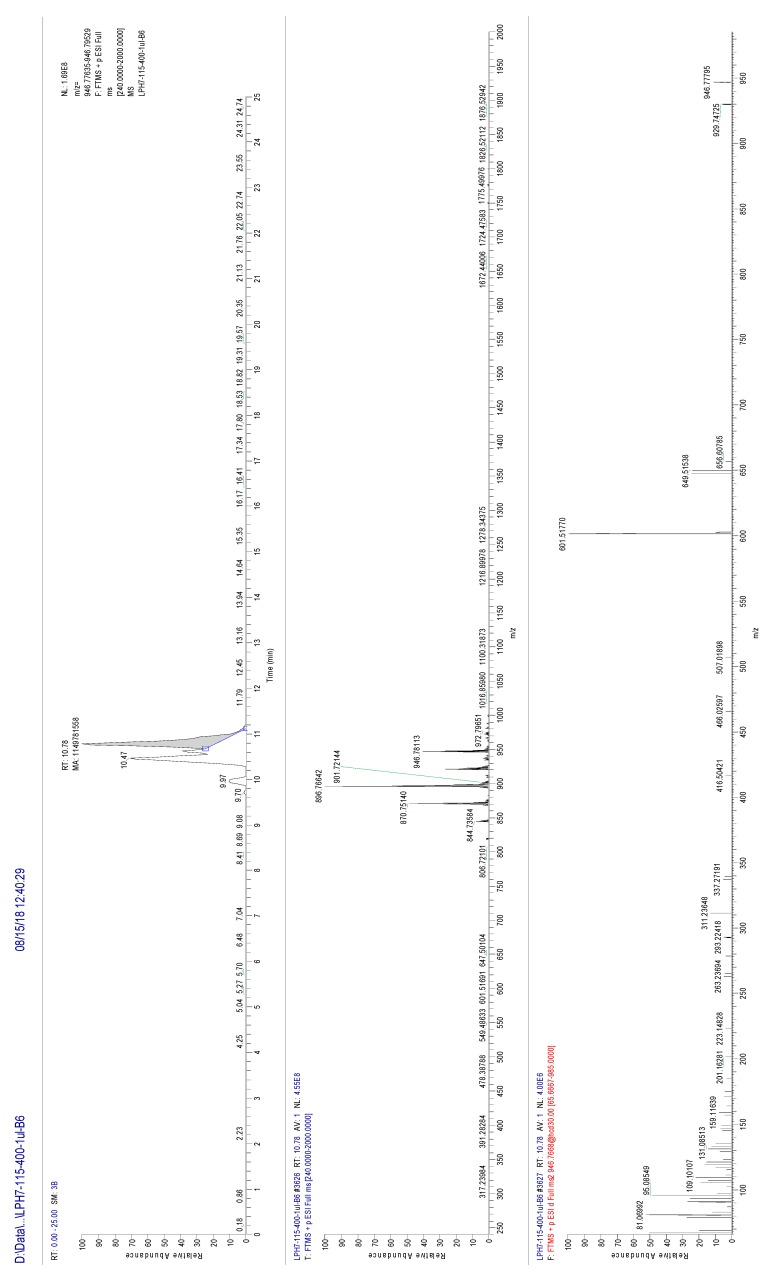
MS and MS/MS spectra of triglyceride (TG (18:1/18:2/22:6)).

**Figure 5 molecules-24-03983-f005:**
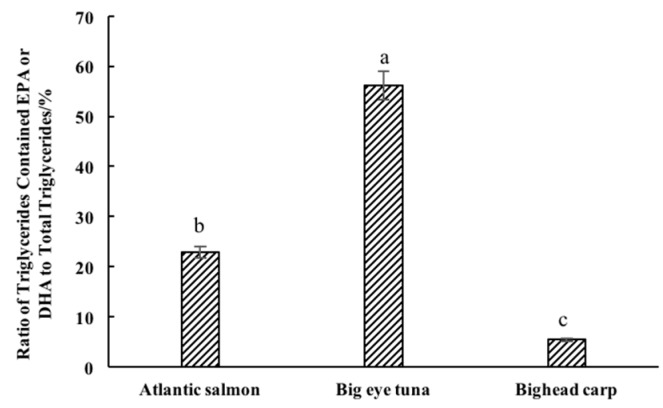
Ratio of triglyceride (TG) molecules contained EPA or DHA to total TG molecules. ^a–c^ means significant difference, *p* < 0.05.

**Figure 6 molecules-24-03983-f006:**
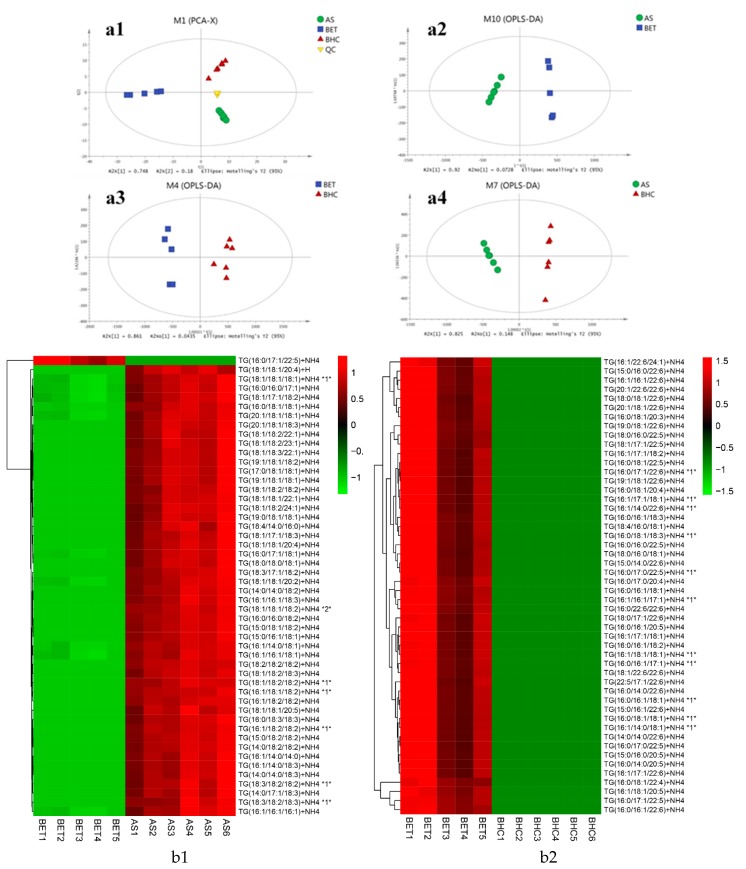
Multivariate statistical analysis and TOP-50 differential TG molecules of big eye tuna (BET), Atlantic salmon (AS) and bighead carp (BHC) heads. (**a1**) PCA (principal component analysis) of three kinds of fish heads; OPLS-DA (orthogonal projection to latent structures-discriminant analysis) of (**a2**) big eye tuna and Atlantic salmon; (**a3**) big eye tuna and bighead carp; (**a4**) Atlantic salmon and bighead carp; Top-50 differential TG molecules of (**b1**) big eye tuna and Atlantic salmon (**b2**) big eye tuna and bighead carp (**b3**) Atlantic salmon and bighead carp heads.

**Table 1 molecules-24-03983-t001:** The lipids contents and determination of TG fraction in three kinds of fish heads. (%, mean ± S.D., *n* = 3).

Samples	Total Lipids	TG	Assay of TG Fraction
Big eye tuna	12.87 ± 4.92 ^b^	85.69 ± 10.22 ^a^	92.22 ± 0.59 ^b^
Atlantic salmon	15.67 ± 2.01 ^a^	60.58 ± 8.32 ^c^	98.52 ± 0.39 ^a^
Bighead carp	5.61 ± 2.08 ^c^	73.39 ± 8.90 ^b^	97.6 ± 0.68 ^a^

^a–c^ values in the same column with different lower-case letters are significantly different at *p* < 0.05.

**Table 2 molecules-24-03983-t002:** Common triglycerides of the lipids recovered from big eye tuna, Atlantic salmon and bighead carp heads (μg/g tissue).

Measured *m*/*z*	Molecular Species	Big Eye Tuna	Atlantic Salmon	Bighead Carp
796.73887	TG (16:0/14:0/16:0)	1.48 ± 0.37 ^a^	0.07 ± 0.03 ^c^	0.55 ± 0.17 ^b^
794.72322	TG (16:0/14:0/16:1)	1.52 ± 0.33 ^a^	1.07 ± 0.08 ^a^	1.54 ± 0.48 ^a^
808.73887	TG (16:0/14:0/17:1)	1.61 ± 0.34 ^a^	0.43 ± 0.04 ^b^	1.35 ± 0.41 ^a^
822.75452	TG (16:0/14:0/18:1)	6.09 ± 1.34 ^a^	2.07 ± 0.16 ^b^	2.69 ± 0.84 ^b^
824.77017	TG (16:0/16:0/16:0)	1.68 ± 0.42 ^a^	0.14 ± 0.02 ^c^	0.56 ± 0.18 ^b^
836.77017	TG (16:0/16:0/17:1)	0.17 ± 0.11 ^c^	1.48 ± 0.16 ^b^	2.39 ± 0.77 ^a^
850.78582	TG (16:0/16:0/18:1)	10.54 ± 2.64 ^a^	4.51 ± 0.41 ^b^	3.33 ± 1.04 ^c^
896.77017	TG (16:0/16:0/22:6)	14.6 ± 3.42 ^a^	2.36 ± 0.16 ^b^	0.48 ± 0.15 ^c^
862.78582	TG (16:0/17:1/18:1)	9.3 ± 2.46 ^a^	4.81 ± 0.51 ^b^	2.94 ± 0.94 ^c^
876.80147	TG (16:0/18:1/18:1)	1.23 ± 0.39 ^c^	18.91 ± 1.66 ^a^	4.68 ± 1.47 ^b^
922.78582	TG (16:0/18:1/22:6)	20.25 ± 19.13 ^a^	5.75 ± 0.43 ^b^	1.38 ± 0.61 ^c^
960.89537	TG (16:0/18:1/24:1)	1.88 ± 1.19 ^b^	4.86 ± 0.65 ^a^	0.34 ± 0.13 ^c^
806.72322	TG (16:1/14:0/17:1)	0.61 ± 0.21 ^b^	0.32 ± 0.04 ^b^	1.04 ± 0.32 ^a^
820.73887	TG (16:1/14:0/18:1)	0.49 ± 0.13 ^b^	3.75 ± 0.26 ^a^	3.47 ± 1.08 ^a^
818.72322	TG (16:1/16:1/16:1)	1.13 ± 0.91 ^c^	3.52 ± 0.41 ^a^	2.71 ± 0.87 ^b^
832.73887	TG (16:1/16:1/17:1)	2.4 ± 2.78 ^a^	0.18 ± 0.02 ^c^	1.55 ± 0.47 ^b^
846.75452	TG (16:1/16:1/18:1)	1.74 ± 1.33 ^b^	0.96 ± 0.1 ^c^	4.38 ± 1.42 ^a^
874.78582	TG (16:1/18:1/18:1)	1.88 ± 2.95 ^c^	25.83 ± 2.31 ^a^	4.35 ± 1.41 ^b^
904.83277	TG (18:0/18:1/18:1)	5.31 ± 3.31 ^b^	15.89 ± 1.5 ^a^	2.31 ± 0.77 ^c^
888.80147	TG (18:1/17:1/18:1)	0.41 ± 0.13 ^c^	5.62 ± 0.63 ^a^	1.47 ± 0.48 ^b^
886.78582	TG (18:1/17:1/18:2)	4.98 ± 1.47 ^a^	3.76 ± 0.37 ^a^	1.33 ± 0.43 ^b^
944.86407	TG (18:1/17:1/22:1)	0.79 ± 0.21 ^b^	1.31 ± 0.18 ^a^	0.23 ± 0.08 ^c^
934.78582	TG (18:1/17:1/22:6)	0.35 ± 0.11 ^a^	0.45 ± 0.06 ^a^	0.32 ± 0.15 ^a^
902.81712	TG (18:1/18:1/18:1)	4.26 ± 3.85 ^b^	34.54 ± 3.07 ^a^	2.42 ± 0.76 ^b^
928.83277	TG (18:1/18:1/20:2)	0.61 ± 0.33 ^b^	7.12 ± 0.71 ^a^	0.88 ± 0.28 ^b^
950.81712	TG (18:1/18:1/22:5)	3.64 ± 1.88 ^b^	13.17 ± 1.56 ^a^	0.52 ± 0.18 ^c^
948.80147	TG (18:1/18:1/22:6)	8.92 ± 7.94 ^a^	6.41 ± 0.47 ^a^	0.85 ± 0.29 ^b^
986.91102	TG (18:1/18:1/24:1)	1.16 ± 0.96 ^b^	5.65 ± 0.71 ^a^	0.11 ± 0.04 ^c^
946.78582	TG (18:1/18:2/22:6)	9.33 ± 3.78 ^a^	5.14 ± 0.65 ^b^	0.8 ± 0.38 ^c^

^a–c^ values in the same line with different lower-case letters are significantly different at *p* < 0.05.

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
