# Peer review of "Comparison of the Fatty Acid and Triglyceride Profiles of Big Eye Tuna (Thunnus obesus), Atlantic salmon (Salmo salar) and Bighead Carp (Aristichthysnobilis) Heads"

_molecules, 2019, doi:10.3390/molecules24213983_

Round 1

Reviewer 1 Report

Line 31  de novo  ………………… italics

Line 87 check the title of the 2s table

230-236 It is important to include the date of capture of each species. It is possible that the annual season generates variations in the content and profile of fatty acids (including: TG, EPA, DHA)?

I am also missing a more detailed discussion of the results for capture of big head carp, the capture in another Chinese lake may influence the content and profile of fatty acids (TG, EPA, DHA…)  

Author Response

Thank you for your comments concerning our manuscript. We have studied all comments carefully and have made correction which we hope meet with their approval. There are some details that need to explain particularly.

All the changes were highlight in red color.

Point 1: Line 31 de novo  ………………… italics

Line 87 check the title of the 2s table

Response 1: Thanks for the comments. We have revised the manuscript according to your comments. (Line 31 and supplementary materials file)

Point 2: 230-236 It is important to include the date of capture of each species. It is possible that the annual season generates variations in the content and profile of fatty acids (including: TG, EPA, DHA)?

Response 2: Thanks for the comments. Fishing season is very likely affecting the quality in some fish species. In this study, all samples were from stationary season. The big eye tuna was captured from Pacific-Indian Ocean in September. We have added the information in Line 241 in the revised manuscript. The farmed Atlantic salmon was from Danish Faroe Islands in October, and the farmed bighead carp was from Qiandao Lake in Chinese Zhejiang region in October. (Line 242-244)

Point 3: I am also missing a more detailed discussion of the results for capture of big head carp, the capture in another Chinese lake may influence the content and profile of fatty acids (TG, EPA, DHA…) 

Response 2: Thanks for the comments, location is very likely affecting the quality in some fish species. In this study, the farmed bighead carp was from Qiandao Lake in Chinese Zhejiang region in October (Line 244). The water quality of Qiandao Lake is top level in China, and it is suitable for bighead carp culture.

Besides that, the bighead carp sample of this study is farmed species not wild species. Because the culturing technology of bighead carp is properly mature, the specific culturing condition could maintain the stable quality and nutrition of bighead carp.

Reviewer 2 Report

The manuscript authored by Zhang et al entitled "Comparison of fatty acid and triglyceride profile among big eye tuna (Thunnusobesus), Atlantic salmon (Salmo salar) and bighead carp (Aristichthysnobilis) heads" is clear, well written, precise and easy to follow.

Authors have investigated the fatty acid and triglyceride composition from big eye tuna, Atlantic salmon and bighead carp by GC-FID and LC-MS/MS. Classes and content of lipids, including fatty acid profiles were determined.

The heads of these fish species have been used in typical Chinese dishes and therefore authors decided to investigate their lipid quality, composition and content. Based on authors’ findings, the head of these fish species are important source of lipids and fatty acids. However, no comparison was carried out with more samples, considering local and seasonal variables. Lipids and its fatty acids can vary a lot in fish seasonally and locally, especially in marine and freshwater environments. This comparison is not clear and should be added as extra experiments.

UPLC Orbitrap was used to characterize TG, quite powerful mass spec analyzer. I wonder if authors considered other lipids and their composition in the head of fish such as brain lipids like phosphatidylcholine (PC), phosphatidylethanolamine (PE), cerebrosides and sulphatides.

Lipid and fatty acid from head of fish as well as their biological importance were poorly discussed. Authors should explore these topics and correlated with other sources of lipids from other kind of food, including exotic dishes.

In summary, quite simple experiments were involved in these studies; however, the work is solid. References were correctly selected and cited along the text.

Discussion fails in giving more support to authors’ conclusion. To finish this revision, besides these above mentioned suggestions, the manuscript is good. However, the manuscript needs more details as recommended herein before its final publication.

Author Response

Thank you for your comments concerning our manuscript. We have studied all comments carefully and have made correction which we hope meet with their approval. There are some details that need to explain particularly.

All the changes were highlight in red color.

Point 1: The heads of these fish species have been used in typical Chinese dishes and therefore authors decided to investigate their lipid quality, composition and content. Based on authors’ findings, the head of these fish species are important source of lipids and fatty acids. However, no comparison was carried out with more samples, considering local and seasonal variables. Lipids and its fatty acids can vary a lot in fish seasonally and locally, especially in marine and freshwater environments. This comparison is not clear and should be added as extra experiments.

Response 1: Thanks for the comments. Fishing season and location indeed affect the quality in some fish species. In this study, all samples were from stationary season and location.

Firstly, big eye tuna is a migratory fish species with specific distribution and short fishing period (June to September every year). The big eye tuna used in this study was captured from Pacific-Indian Ocean in September (Line 241).

Secondly, the Atlantic salmon and bighead carp of this study are two farmed fish species not wild species. Nowadays, the culturing technologies of the two fish species are properly mature, the specific culturing condition could maintain the stable quality and nutrition. In this study, the farmed Atlantic salmon was from Danish Faroe Islands in October, which is most suitable for the Atlantic salmon. The farmed bighead carp was from Qiandao Lake in Chinese Zhejiang region in October, the water quality of Qiandao Lake is top level in China, and it is suitable for bighead carp culture. (Line 242-244)

Thirdly, this study is aimed at comparing different marine and freshwater species from their fatty acid and triglyceride profile. The seasonal difference of same marine or freshwater fish species could be investigated in future study.

Thus we did not compared seasonal variables. Thanks again for comments.

Point 2: UPLC Orbitrap was used to characterize TG, quite powerful mass spec analyzer. I wonder if authors considered other lipids and their composition in the head of fish such as brain lipids like phosphatidylcholine (PC), phosphatidylethanolamine (PE), cerebrosides and sulphatides.

Response 2: Thanks for the comments. TG is the major components of fish oil, and it is important to human body health. Nowadays, the fish oil products are mainly ethyl ester (EE) type. It has reported that TG type n-3 long chain fatty acids (especially EPA and DHA) have better bioavailability than EE type. In this study, we found that the three fish heads are rich in TG type EPA/DHA. Therefore, we were mainly focused on TG in this study (Line 132-139). Investigations on phospholipids (including PC, PE PI and so on) of the three fish heads could be carried out in our further study.

Point 3: Lipid and fatty acid from head of fish as well as their biological importance were poorly discussed. Authors should explore these topics and correlated with other sources of lipids from other kind of food, including exotic dishes.

Discussion fails in giving more support to authors’ conclusion.

Response 3: Thanks for the comments. The results of fatty acid profile had already furtherly discussed. It was mainly focused on the biological importance of the dominant fatty acid and the comparison with other fish species. (Line 86-105)

The original text is as follows,

As showed in Figure 1A, the bighead carp head showed a significantly higher (P<0.05) value of monounsaturated fatty acids (MUFA) (14.33%, 46.78% and 61.81% in big eye tuna, Atlantic salmon and big head carp heads, respectively, the same below), while two kinds of marine fish heads contained significantly higher (P<0.05) amount of PUFA than bigheads carp head (70.5%, 43.33% and 23.82%). Compared with MUFA and PUFA, saturated fatty acid (SFA) was the fewest fraction in all three fish heads (15.16%, 9.37% and 14.37%).

Palmitic (16:0) was the major SFA in all three fish head, the consumption of palmitic is related to the risk of cardiovascular disease. Palmitoleic (16:1) was the major MUFA in all three fish head which found in human skin and decreases with age [21]. In this study, palmitic was accounted for 32.53%, 57.33% and 31.84% in SFA, and palmitoleic was 29.85%, 52.19% and 67.87% in MUFA. Large yellow croaker is a kind of typical marine fish species, its muscle has 57.77% palmitic in SFA, and 22.88% palmitoleic in MUFA [22]. Meanwhile, tilapia is another major freshwater fish species. It was reported that 16:0 and 16:1 also are the predominant SFA and MUFA in tilapia head, and they were accounted for 68.8% and 13.78% in SFA and MUFA, respectively [23].

EPA and DHA are two representative n-3 PUFA that are widely exist in marine triglyceride, phospholipids, ethyl ester and so on [24]. There are larger number of researches have demonstrated that dietary EPA and DHA could improve brain function, inhibit the activity of tumor and regulate lipid and glucose metabolism [25-28]. In this study, the amounts of EPA+DHA accounted for about 91.18, 34.42 and 53.18% in PUFA of three kinds of fish heads. In contrast, the amount of EPA+DHA was 52.45% and 4.24% of PUFA in muscle of large yellow croaker and head of tilapia [22,23].

Reviewer 3 Report

This paper has a great potential to be accepted, but some important issues have to be clarified or fixed before a positive realization. I have many doubts about the style of this paper. My concerns are mainly about how manuscript has been organized. The authors did not separate well results from discussion, this makes a quite confusion to follow different aspect of this interesting work. There are written parts and figures from discussion that should be transfer to the results section and vice versa. Results presented with contemporary written discussion would give a better final aspect, more concise and much more undeviating. I suggest to put together the results and discussion, not to distinct them. This is because it is quite difficult to present and then to explain and comment some important findings. Especially when the appearance of some TGs, their fragmentation and finally putative identification are concerned.

Here summarize some of important issues to consider:

Line 48. Please eliminate the part “Triglycerides (TG) are the most abundant lipids in the human body” – it is well known fact.

Lines 54-55. Please re-write the following sentence “In general, a molecule with a short fatty acid chain peaks before a long-chain molecule, and a molecule with a high degree of unsaturation precedes a peak with a high saturation molecule”. Instead of “peaks” it is better to use the term “eluate” .

Line 71 and line 92. You did UHPLC - Q Exactive Orbitrap  not simple LC-MS/MS, please correct: Also, I would suggest the following abbreviation for the acquisition mode you applied: High resolution mass spectrometry data dependent analysis (HRMS-DDA)

Figure 1. The quality of the TIC presentation can be improved using Xcalubur facilities. The chromatograms are very descriptive regarding the TG content by they are not presented well.

Table 3S. Please put just the TG abbreviation (without +NH4) in the first column. In the second column instead of m/z please put adduct (M+NH4)+. The exact mass must be presented with at least 4 (even 5) decimals not with just two. This is what Q-Exactive is supposed to perform highly accurate masses.

Table 2: Also here put adduct (M+NH4) and 4-5 decimals.

Line 86-88 Please elaborate better. There are no comments on this issue. Something is wrong with Table 1S the title does not correspond with content. The tables 1S and 2S present very interesting results but they are not clear. Please rearrange them in the following way: Fatty acid, Big eye tuna, Atlantic salmon and Big head carp.

Figures 4. The quality of the chromatogram is questionable, resolution is low, it has to be much better. Fragmentation pattern from the figure 4. It is not possible to attach covalently the anion NH4+ to the TG as you put in the fragmentation scheme. Please present in appropriate way.

Please explain why you used ng/μL to present the quantity of TGs? I suggest authors to present the final concentration as μg or mg per tissue (mg or g)   

Author Response

Thank you for your comments concerning our manuscript. We have studied all comments carefully and have made correction which we hope meet with their approval. There are some details that need to explain particularly.

All the changes were highlight in red color.

Point 1: This paper has a great potential to be accepted, but some important issues have to be clarified or fixed before a positive realization. I have many doubts about the style of this paper. My concerns are mainly about how manuscript has been organized. The authors did not separate well results from discussion, this makes a quite confusion to follow different aspect of this interesting work. There are written parts and figures from discussion that should be transfer to the results section and vice versa. Results presented with contemporary written discussion would give a better final aspect, more concise and much more undeviating. I suggest to put together the results and discussion, not to distinct them. This is because it is quite difficult to present and then to explain and comment some important findings. Especially when the appearance of some TGs, their fragmentation and finally putative identification are concerned.

Response 1: Thanks for comments. The results and discussion have put together in the revised manuscript. (Line 74)

Point 2: Line 48. Please eliminate the part “Triglycerides (TG) are the most abundant lipids in the human body” – it is well known fact.

Lines 54-55. Please re-write the following sentence “In general, a molecule with a short fatty acid chain peaks before a long-chain molecule, and a molecule with a high degree of unsaturation precedes a peak with a high saturation molecule”. Instead of “peaks” it is better to use the term “eluate”.

Line 71 and line 92. You did UHPLC - Q Exactive Orbitrap not simple LC-MS/MS, please correct: Also, I would suggest the following abbreviation for the acquisition mode you applied: High resolution mass spectrometry data dependent analysis (HRMS-DDA)

Response 2: Thanks for the comments. We have revised the manuscript according to your comments. (Line 48, 53, 54, 70, 158, 160).

Point 3: Figure 1. The quality of the TIC presentation can be improved using Xcalubur facilities. The chromatograms are very descriptive regarding the TG content by they are not presented well.

Response 3: Thanks for comments. we have provided high-resolution chromatogram figure. (Line 142)

Point 4: Table 3S. Please put just the TG abbreviation (without +NH4) in the first column. In the second column instead of m/z please put adduct (M+NH4)+. The exact mass must be presented with at least 4 (even 5) decimals not with just two. This is what Q-Exactive is supposed to perform highly accurate masses.

Table 2: Also here put adduct (M+NH4) and 4-5 decimals.

Response 4: Thanks for comments. We have revised the manuscript according to your comments. (Table 3S, Table 2)

Point 5: Line 86-88 Please elaborate better. There are no comments on this issue. Something is wrong with Table 1S the title does not correspond with content. The tables 1S and 2S present very interesting results but they are not clear. Please rearrange them in the following way: Fatty acid, Big eye tuna, Atlantic salmon and Big head carp.

Response 5: Thanks for comments. The mistake of title of Table 1S was corrected, and the Table 1S and 2S were rearranged as your recommend way. The results of fatty acid profile have already furtherly discussed, which is mainly focused on the biological importance of the dominant fatty acid and the comparison with other fish species. (Line 86-105).

The original text is as follows,

As showed in Figure 1A, the bighead carp head showed a significantly higher (P<0.05) value of monounsaturated fatty acids (MUFA) (14.33%, 46.78% and 61.81% in big eye tuna, Atlantic salmon and bighead carp heads, respectively, the same below), while two kinds of marine fish heads contained significantly higher (P<0.05) amount of PUFA than bigheads carp head (70.5%, 43.33% and 23.82%). Compared with MUFA and PUFA, saturated fatty acid (SFA) was the fewest fraction in all three fish heads (15.16%, 9.37% and 14.37%).

Palmitic (16:0) was the major SFA in all three fish head, the consumption of palmitic is related to the risk of cardiovascular disease. Palmitoleic (16:1) was the major MUFA in all three fish head which found in human skin and decreases with age [21]. In this study, palmitic was accounted for 32.53%, 57.33% and 31.84% in SFA, and palmitoleic was 29.85%, 52.19% and 67.87% in MUFA. Large yellow croaker is a kind of typical marine fish species, its muscle has 57.77% palmitic in SFA, and 22.88% palmitoleic in MUFA [22]. Meanwhile, tilapia is another major freshwater fish species. It was reported that 16:0 and 16:1 also are the predominant SFA and MUFA in tilapia head, and they were accounted for 68.8% and 13.78% in SFA and MUFA, respectively [23].

EPA and DHA are two representative n-3 PUFA that are widely exist in marine triglyceride, phospholipids, ethyl ester and so on [24]. There are larger number of researches have demonstrated that dietary EPA and DHA could improve brain function, inhibit the activity of tumor and regulate lipid and glucose metabolism [25-28]. In this study, the amounts of EPA+DHA accounted for about 91.18, 34.42 and 53.18% in PUFA of three kinds of fish heads. In contrast, the amount of EPA+DHA was 52.45% and 4.24% of PUFA in muscle of large yellow croaker and head of tilapia [22,23].

Point 6: Figures 4. The quality of the chromatogram is questionable, resolution is low, it has to be much better. Fragmentation pattern from the figure 4. It is not possible to attach covalently the anion NH4+ to the TG as you put in the fragmentation scheme. Please present in appropriate way.

Response 6: Figure 4 showed the MS and MS/MS chromatogram of the molecule TG (18:1/18:2/22:6). We put a high-resolution figure instead of the old one. Meanwhile, the m/z information was indicated in line 140, 143-144. Perhaps this is more proper expression. Thanks for your careful comments.

Point 7: Please explain why you used ng/μL to present the quantity of TGs? I suggest authors to present the final concentration as μg or mg per tissue (mg or g)   

Response 7: Thanks for your kind suggestion. The quantitative method was introduced in Line 324-333. Now the results have already converted to μg/(g tissue) (Table 2, supplementary materials). Thanks again for your very valuable comments.

Round 2

Reviewer 2 Report

Authors have answered, commented and improved this new version of the manuscript. They considered the most important queries that I had asked. Therefore, in the present form, I recommended it for publication.